# Relationship between Land Use and Spatial Variability of Atmospheric Brown Carbon and Black Carbon Aerosols in Amazonia

Fernando G. Morais [1,2,*], Marco A. Franco [1,3,*], Rafael Palácios [4], Luiz A. T. Machado [1,3], Luciana V. Rizzo [1,3], Henrique M. J. Barbosa [1,5], Fabio Jorge [1], Joel S. Schafer [6], Brent N. Holben [6], Eduardo Landulfo [2] and Paulo Artaxo [1,3]

1 Institute of Physics, University of São Paulo, Rua do Matão, São Paulo 05508-090, Brazil
2 Centro de Lasers e Aplicações (CELAP), Instituto de Pesquisas Energéticas e Nucleares (IPEN), Av. Prof. Lineu Prestes 2242, São Paulo 05508-000, Brazil
3 Research Centre for Greenhouse Gas Innovation (RCGI-POLI), University of São Paulo, Av. Professor Mello Moraes, 2231, Butantã, São Paulo 05508-030, Brazil
4 Faculdade de Meteorologia, Instituto de Geociências, Universidade Federal do Pará, Belém 66075-110, Brazil
5 Physics Department, University of Maryland Baltimore County, Baltimore, MD 21250, USA
6 NASA Goddard Space Flight Center (GSFC), Greenbelt, MD 20771, USA
* Correspondence: fmorais@usp.br (F.G.M.); marco.franco@usp.br (M.A.F.)

**Abstract:** The aerosol radiative effect is an important source of uncertainty in estimating the anthropogenic impact of global climate change. One of the main open questions is the role of radiation absorption by aerosols and its relation to land use worldwide, particularly in the Amazon Rainforest. Using AERONET (Aerosol Robotic Network) long-term measurements of aerosol optical depth (AOD) at a wavelength of 500 nm and absorption AOD (AAOD) at wavelengths of 440, 675, and 870 nm, we estimated the fraction and seasonality of the black carbon (BC) and brown carbon (BrC) contributions to absorption at 440 nm. This was conducted at six Amazonian sites, from central Amazon (Manaus and the Amazon Tall Tower Observatory—ATTO) to the deforestation arc (Rio Branco, Cuiabá, Ji-Paraná, and Alta Floresta). In addition, land use and cover data from the MapBiomas collection 6.0 was used to access the land transformation from forest to agricultural areas on each site. The results showed, for the first time, important geographical and seasonal variability in the aerosol optical properties, particularly the BC and BrC contributions. We observed a clear separation between dry and wet seasons, with BrC consistently accounting for an average of approximately 12% of the aerosol AAOD at 440 nm in the deforestation arc. In central Amazon, the contribution of BrC was approximately 25%. A direct relationship between the reduction in forests and the increase in the area dedicated to agriculture was detected. Moreover, places with lower fractions of forest had a smaller fraction of BrC, and regions with higher fractions of agricultural areas presented higher fractions of BC. Therefore, significant changes in AOD and AAOD are likely related to land-use transformations and biomass burning emissions, mainly during the dry season. The effects of land use change could introduce differences in the radiative balance in the different Amazonian regions. The analyses presented in this study allow a better understanding of the role of aerosol emissions from the Amazon Rainforest that could have global impacts.

**Keywords:** AERONET; Amazon; brown carbon; black carbon; land use; remote sensing

## 1. Introduction

When evaluating the impact of human activity on global climate change, the aerosol radiative effect is a significant source of uncertainty [1]. One of the main open questions is the role of radiation absorption by aerosols and their impacts on Earth's radiative balance [2]. Global climate models typically underestimate the large-scale radiative forcing of

absorbing aerosols when compared to observations. The hypothesis to explain this discrepancy includes underestimation of emission rates and not accounting for the contribution of light-absorbing organic aerosols, as well as aerosol aging processes that enhance absorption [3]. Aerosol particles in the atmosphere originate from different sources and may be classified into two major composition classes according to their chemical origin: organics and inorganics [4]. Organics dominate the aerosol composition, particularly over forested regions, accounting for approximately 80% of the PM1 mass [5]. In particular, carbonaceous light-absorbing aerosols, which can be both organic and inorganic, are usually referred to as black carbon (BC) and brown carbon (BrC) [6].

BC, which is mainly composed of elemental carbon, is emitted directly into the atmosphere and has unique physical properties, strongly absorbing solar radiation in the visible and infrared spectral regions [7]. The main global sources of BC are the combustion of fossil fuels and biomass burning [8]. Once emitted, BC can be transported on a regional and continental scale and removed from the atmosphere by dry and wet deposition, resulting in an average atmospheric lifetime of approximately one week [9]. In contrast, BrC consists of organic light-absorbing aerosols, absorbing mainly in the ultraviolet spectral range and the shorter wavelengths of the visible range. Parts of BrC may consist of secondary organic aerosols (SOA) formed in the atmosphere by chemical reactions associated with the oxidation of volatile organic compounds (VOCs) [10]. Moreover, BrC can originate from biological processes related to the decomposition of organisms, providing the light-absorbing component of dark soils, for example, [6]. Along with SOA and biological processes, a fraction of BrC can also be emitted from biomass burning and fuel combustion [11–13]. As BC and BrC absorb in different regions of the solar spectrum, multi-wavelength absorption measurements can, in principle, be used to distinguish their relative contributions.

However, the characterization of aerosol spectral properties is challenging due to seasonal and spatial variations in concentrations, chemical composition, and physical and optical properties. It requires a joint effort to measure these properties in different regions [14]. Due to the need for consistent regional and global measurements, the National Aeronautics and Space Administration (NASA) established a global network of sun photometers instruments for monitoring aerosol properties, the AERONET (Aerosol Robotic Network) [15]. Several studies have addressed different aspects of aerosol physical properties worldwide using AERONET measurements [16,17].

This study aims to characterize the long-term spatiotemporal variability of aerosol optical properties, specifically looking into the BC and BrC contributions to aerosol absorption at six different Amazonian sites and describing possible associations with land use change in the Amazon. We focus on this region because of its climatic relevance [18] and unique conditions to investigate the seasonal behavior of aerosol properties [19]. The sites are in areas with different levels of anthropogenic influence in Amazonia, from natural forest conditions to agricultural lands heavily impacted by biomass burning emissions. Here, we use AERONET's long-term measurements in the Amazon, covering more than 23 years at some sites. In addition, we analyzed and characterized the land use evolution over the years for each site, describing, for the first time, the land transformation from forest to agricultural regions and its relationship to BrC and BC fractions in the Amazonian atmosphere.

## 2. Materials and Methods

Aerosol optical depth (AOD) at 500 nm and the absorption AOD (AAOD) at 440, 675, and 870 nm measured in six different Amazonian sites, together with land use cover obtained by the MapBiomas platform [20], were used. The separation between BC and BrC was obtained according to the methodology developed by Wang et al., 2016 [21]. Ångström matrices based on Cazorla et al., 2013 [22] were used to investigate the aerosol carbon mixture for each site, which allowed the discrimination of the important processes and mechanisms that play the most significant role for each Brazilian Amazon AERONET site.

We analyzed the AOD and the AAOD from photometers of the AERONET network, particularly focusing on the absorption component. The AERONET network is a global partnership for monitoring aerosols by ground-based remote sensing, maintained by NASA's EOS (Earth Observing System) and expanded by collaborators from national agencies, institutes, universities, and other sponsors. The network comprises automatic solar and lunar spectral radiometers, models CIMEL Electronic 318A and 318T, whose measurements allow near real-time monitoring of parameters, such as AOD, AAOD, and other inversion-derived properties. These include the aerosol size distribution, and single scattering albedo, among others. These products can be freely accessed online (https://aeronet.gsfc.nasa.gov/ (accessed on 1 June 2022)), where additional information about the network and processing algorithms are available.

The AERONET network has a very consistent data series in the Amazon, where some sites have a time span of more than 20 years. We used the following sites in this study: Alta Floresta (1999–2021), Cuiabá (2001–2021), Ji-Paraná (2006–2021), and Rio Branco (2000–2021), which comprehend the sites located along the deforestation arc [23], and Manaus (2011–2019) and the Amazon Tall Tower Observatory (ATTO, 2016–2021) sites, located in relatively clean and pristine sites in central Amazon. It is worth mentioning that the AERONET sun photometers in Manaus and at the ATTO site are located outside the urban region, with the one in Manaus being located upwind of the city, i.e., in a place where, in general, little urban pollution affects the region due to the wind direction. AERONET sites in the deforestation arc are present in relatively small cities, where the urban contribution is less significant than, for example, local and regional fires. Palácios et al. (2020) describe the referred AERONET site locations in more detail [24]. AERONET data quality level 1.5 was used in this study due to its consistent data coverage for all sites [25].

The AAOD data was used to calculate the fraction of BC and BrC for each site. To obtain the contribution of BrC through AERONET's absorption measurements, we applied a methodology [21] using the information from the wavelength-dependent measurements. The wavelength dependence of aerosol absorption is quantified by the absorption Ångström exponent (AAE), considering absorption measurements at two wavelengths, $\lambda_1$ and $\lambda_2$, according to Equation (1).

$$AAE_{\lambda 1 - \lambda 2} = -\{\ln [AAOD(\lambda 1)/AAOD(\lambda 2)]\}/[\ln (\lambda 1/\lambda 2)], \qquad (1)$$

BrC typically shows greater AAE values when compared to BC, and this characteristic was used to distinguish their contribution to aerosol absorption. The method considers the AAOD at 440 nm as composed of the sum of the contributions of AAOD BC and AAOD BrC, according to Equation (2).

$$AAOD_{440\ nm} = AAOD\ BC_{440\ nm} + AAOD\ BrC_{440\ nm}. \qquad (2)$$

The calculation consists of first obtaining the contribution of BC absorption to the total AAOD and then retrieving the contribution of BrC. The AAE of AAOD was calculated considering three selected wavelengths: 440 nm, where BrC interacts more efficiently; the visible red wavelength at 675 nm, and 870 nm, where BC absorption dominates. Due to the higher absorption efficiency of BrC at shorter wavelengths, the AAE is not constant throughout the spectrum, showing a wavelength dependency itself. The wavelength dependence of AAE (WDA) is defined as Equation (3).

$$WDA = AAE_{440\text{-}870\ nm} - AAE_{675\text{-}870\ nm}. \qquad (3)$$

Based on Mie theory, theoretical values for BC WDA were obtained by considering polydisperse coated BC particles with different size distributions. BC was assumed to be composed of a core of internally mixed monodisperse BC with a refractive index of 1.95–0.76 i [8] and a coating with a refractive index of 1.55–0.001 i. The BC density was assumed to be 1.8 g cm$^{-3}$ [26].

To calculate the BC contribution to absorption, we isolate the term $AAE_{440-870\ nm}$ in Equation (3) and use Equation (2). We obtain Equations (4) and (5).

$$BC\ AAE_{440\text{-}870\ nm} = BC\ WDA + AAE_{675\text{-}870\ nm}, \tag{4}$$

$$BC\ AAOD_{440\ nm} = AAOD_{870\ nm} \times (440/870) - {}^{BC\ AAE\ 440\text{-}870\ nm} \tag{5}$$

Therefore, applying Equation (5) to Equation (1), we derive the contribution of BrC to light absorption as described in Equation (6).

$$AAOD\ BrC_{440\ nm} = AAOD_{440\ nm} - AAOD\ BC_{440\ nm}. \tag{6}$$

The above method estimates the BC and BrC fractions from the total AAOD. Uncertainties are discussed in detail in the literature [21,27,28].

The Ångström matrix was used to investigate aerosol sources and their carbonaceous mixtures. Aerosols classified by the Ångström matrix [22,25,29] use the AAE values as indicators of the aerosol chemical nature and the scattering Ångström exponent (SAE) as an indication of the average particle size distribution. The methodology gives hints about the aerosol mixture and its likely origin at a particular site. Results from the Ångström matrix should be interpreted as a qualitative analysis. This study applied the same methodology as Cazorla et al., 2013 [22] for different Amazonian regions associated with different types of aerosol populations. It is worth mentioning that the method depends exclusively on the SAE and AAE, which are intensive aerosol properties, meaning that no correctness in using this method is required.

In addition to AERONET, the MapBiomas platform (https://mapbiomas.org/ (accessed on 1 June 2022)) was used as a source of land-use data for the different Amazonian sites. In particular, the facility allows the user to obtain different classes of land use and cover from 1985 to 2020, discriminating the regions by features, such as biomes, urban areas, and indigenous reserves. The catalog is generated based on remote sensing data and machine learning, creating annual maps of land use [20]. To have a comprehensive picture of the land use in the selected regions, we evaluated the percentages of forested, agricultural, and non-forested areas from the latest MapBiomas data collection 6.0.

## 3. Results and Discussion

### 3.1. The Contribution of BC and BrC to the Aerosol Optical Properties

Figure 1 shows the strong seasonality of the AOD, which reflects the variability of the Amazonian aerosol emissions, i.e., the alternation of rainy and biomass burning periods. Based on the observed seasonality, we divided the observation period into wet (January–June) and dry (July–December) seasons [30]. Table 1 presents the average AOD at 500 nm and total AAOD at 440 nm, respectively, for each season.

Lower AOD values were observed during the wet season, ranging from 0.08 to 0.13. This is mainly due to the wet deposition, but also to the nature of the aerosol sources, which are related to natural emissions (primary and secondary organic aerosols—SOA), sporadic long-range transport of aerosol plumes from the African continent (mainly dust), and some local urban emissions. It is worth noting that in the wet season, AODs at sites in the deforestation arc do not differ significantly from those of remote sites. Regional natural emissions and atmospheric aerosol processing play a more critical role in aerosol loading and properties.

There is a substantial increase in the average AOD during the dry season, spanning from 0.19 at the ATTO site to 0.49 in Alta Floresta, which is mainly due to regional biomass burning emissions, as well as long-range transport even from West Africa [5,24]. In particular, the sites in the deforestation arc presented significantly higher average AOD values, from 0.39 to 0.49, with peaks higher than 3.0 during the most intense biomass burning periods. In contrast, in central Amazon, the average AOD varies from 0.19 to 0.26. The substantial year-to-year variability observed in Figure 1 is associated with different

precipitation rates and large-scale meteorological phenomena, such as El Niño and La Niña [31], and governmental policies for managing deforestation. It is worth mentioning that many of those policies have allowed for significant deforestation increases over the last 4–5 years [32,33].

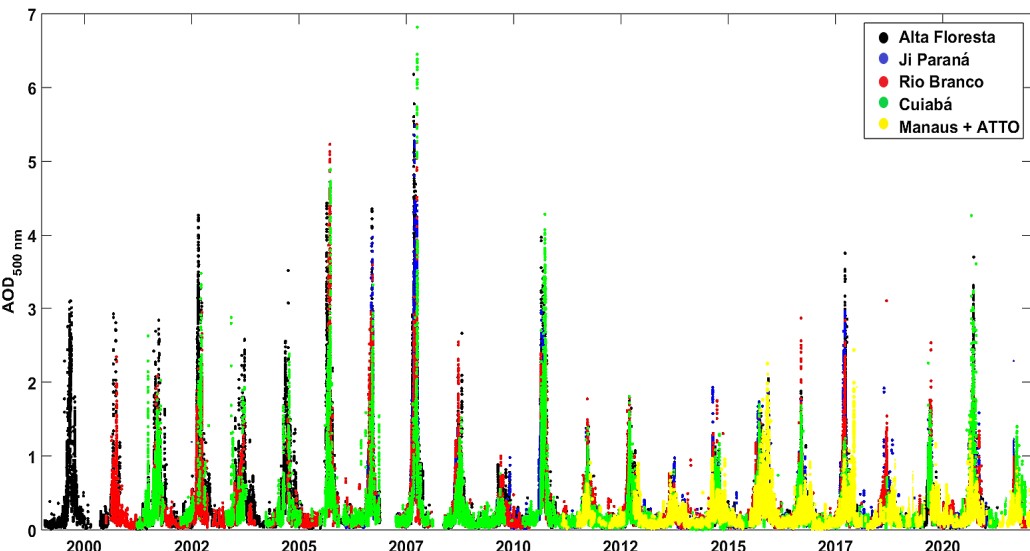

**Figure 1.** Time series from 1999 to 2021 of aerosol optical depth (AOD) at 500 nm at the six Amazonian sites. The strong seasonality is evident from the wet to dry season for all sites.

**Table 1.** Average aerosol optical depth (AOD) and absorption AOD (AAOD) at the wavelengths of 500 nm and 440 nm, respectively, during the wet (January–June) and dry (July–December) seasons for the six Amazonian sites. The period considered for each site is indicated in parenthesis. The uncertainty is the standard deviation.

| Site (Period) | $AOD_{500\ nm,\ wet}$ | $AOD_{500\ nm,\ dry}$ | $AAOD_{440\ nm,\ wet}$ | $AAOD_{440\ nm,\ dry}$ |
|---|---|---|---|---|
| Alta Floresta (1999–2021) | $0.08 \pm 0.06$ | $0.49 \pm 0.64$ | $0.0081 \pm 0.0079$ | $0.035 \pm 0.040$ |
| Ji-Paraná (2006–2021) | $0.08 \pm 0.04$ | $0.43 \pm 0.56$ | $0.0069 \pm 0.0062$ | $0.029 \pm 0.033$ |
| Rio Branco (2000–2021) | $0.10 \pm 0.05$ | $0.39 \pm 0.46$ | $0.0150 \pm 0.0130$ | $0.040 \pm 0.043$ |
| Cuiabá (2001–2021) | $0.10 \pm 0.09$ | $0.39 \pm 0.48$ | $0.0140 \pm 0.0170$ | $0.040 \pm 0.046$ |
| Manaus (2011–2019) | $0.11 \pm 0.07$ | $0.26 \pm 0.21$ | $0.0079 \pm 0.0094$ | $0.015 \pm 0.016$ |
| ATTO (2016–2021) | $0.13 \pm 0.09$ | $0.19 \pm 0.11$ | $0.0066 \pm 0.0098$ | $0.015 \pm 0.018$ |

The AAOD also follows seasonal variation, but differences among the sites stand out. The AAOD values during the wet season ranged from 0.0066 at the ATTO site and Ji-Paraná to 0.015 in Rio Branco. ATTO and Manaus showed a difference in AAOD of 0.0013, which, although low, is likely related to the city's urban emissions (i.e., thermoelectric power plants nearby the site). During the dry season, the AAOD values ranged from 0.015 at the ATTO site and Manaus to 0.040 at Rio Branco and Cuiabá. In particular, the central Amazonian sites did not show differences among them, showing that, despite the urban emissions from Manaus, the impact of regional biomass burning is much more dominant in the AAOD. Sites along the deforestation arc have the highest AAOD values, almost four

times the sites in central Amazonia, since they are the most affected by changes in land use, particularly by the strong biomass burning activities.

All six sites showed clear differences in AOD and total AAOD. The average AOD in Alta Floresta was more than twice the AOD measured at the ATTO site during the dry season. The AAOD in Alta Floresta, Rio Branco, and Cuiabá in the dry season was also between two and three times higher than those measured in Manaus and ATTO. In Brazil, the main land-use conversion tool, which is biomass burning [34], explains the significant differences between central Amazon sites and those located along the deforestation arc. It is worth noting that even during the wet season, Rio Branco and Cuiabá still presented an average AAOD twice as high as the other sites. During the dry season, consistency was maintained, and these sites had the highest average values of AAOD. In contrast, Ji-Paraná presented an interesting behavior: among the sites in the deforestation arc, this was the one with the lowest mean AAOD value at 440 nm.

The BC and BrC AAODs and their respective fractions regarding the total AAOD at 440 nm were retrieved from the measured AAODs at 440, 675, and 870 nm, as described in Section 2 for the six Amazonian sites. The calculations were performed considering all the historical time series of each site. Figure 2 illustrates the time series of the AAOD at 440 nm for the BrC and BC components in Alta Floresta. The data presented gaps mainly during the wet season due to the large number of clouds and rainfall in the Amazon Rainforest. It is worth noting that the lower amount of BC during 2012–2017 compared to the previous period is due to policies for containing deforestation applied in the Amazon, which became more relaxed over time [32,33]. Despite the differences in data coverage for each season, Table 2 shows the average AAODs discriminated for the BC and BrC components at 440 nm and the respective species fraction for both wet and dry seasons.

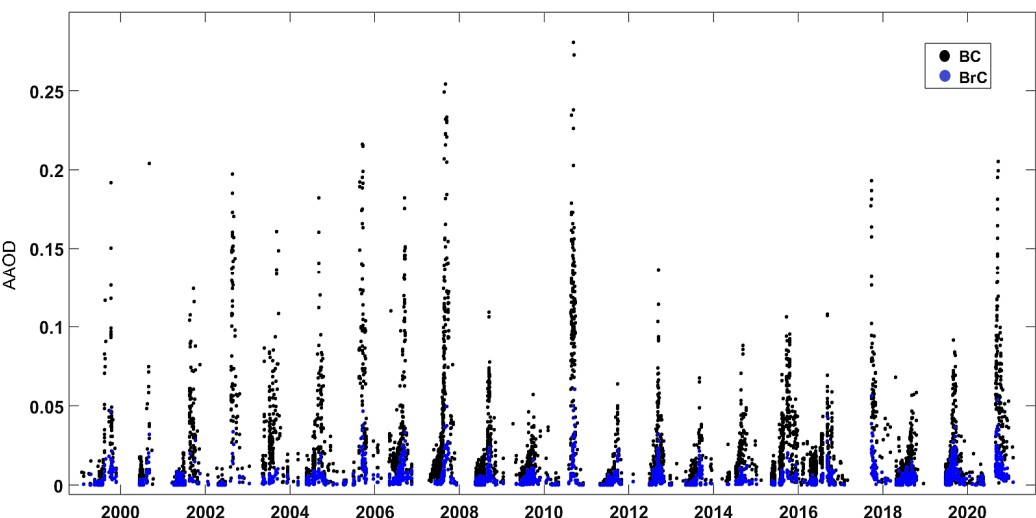

**Figure 2.** The time series of the AAOD at 440 nm for BrC and BC in Alta Floresta.

Both AAOD BC and AAOD BrC present a clear seasonality. The BC component dominates the total AAOD regardless of site and season, which is corroborated by previous studies [8,18,26,27]. The contribution of BrC to AAOD ranges between 9 and 27%. Interestingly, in the wet season, the AAOD BrC at the Rio Branco site is higher than at ATTO and Manaus (0.0033 and 0.0026, respectively). The aerosol population at these forest sites is dominated by SOA, which can be light-absorbing and contribute to AAOD BrC. A possible cause for the high AAOD BrC in Rio Branco is local pollution sources whose aerosol plumes may reach the site from time to time [28].

In contrast, the sites along the deforestation arc were the ones that presented the highest mean AAOD values for BC, which varied between 0.0068 and 0.014 in Ji-Paraná and Cuiabá, respectively. Concerning the central Amazon sites, the average AAODs of BC

were the lowest observed among the six sites, varying between 0.0046 and 0.0055 at ATTO and Manaus, respectively.

**Table 2.** Average AAODs at 440 nm and the BrC and BC components fraction during the wet (January–June) and dry (July–December) seasons. The periods considered for each site are the same as in Table 1.

| Site | AAOD $BrC_{wet}$ % $BrC_{wet}$ | AAOD $BrC_{dry}$ % $BrC_{dry}$ | AAOD $BC_{wet}$ % $BC_{wet}$ | AAOD $BC_{dry}$ % $BC_{dry}$ |
|---|---|---|---|---|
| Alta Floresta | 0.0015 ± 0.0016 10% | 0.0063 ± 0.0071 9% | 0.0073 ± 0.0077 90% | 0.033 ± 0.039 91% |
| Ji-Paraná | 0.0021 ± 0.0021 16% | 0.0053 ± 0.0055 11% | 0.0068 ± 0.0066 84% | 0.028 ± 0.033 89% |
| Rio Branco | 0.0047 ± 0.0042 20% | 0.0080 ± 0.0074 11% | 0.012 ± 0.011 80% | 0.037 ± 0.041 89% |
| Cuiabá | 0.0031 ± 0.0034 13% | 0.0092 ± 0.0097 15% | 0.014 ± 0.018 87% | 0.035 ± 0.040 85% |
| Manaus | 0.0026 ± 0.0045 22% | 0.0058 ± 0.0056 24% | 0.0055 ± 0.0083 78% | 0.012 ± 0.013 76% |
| ATTO | 0.0033 ± 0.0066 27% | 0.0060 ± 0.0075 25% | 0.0046 ± 0.0078 73% | 0.014 ± 0.016 75% |

During the dry season, the mean AAOD values for BrC ranged between 0.0053 in Ji-Paraná and 0.0092 in Cuiabá, respectively. Manaus and ATTO showed similar mean AAOD values for BrC of 0.0058 and 0.0060, respectively. The BC component, in contrast, always had higher averages for sites along the deforestation arc. Generally, the averages varied between 0.012 and 0.014 in Manaus and ATTO, respectively, and the highest values for Rio Branco and Cuiabá were 0.037 and 0.035, respectively. These results indicate that sites heavily impacted by forest-agriculture transformation processes are also the ones with the highest average AAOD values for the BrC components and, mainly, BC.

Regarding the fractions of the contributions of BrC and BC, Table 2 shows that the sites along the deforestation arc have the lowest fractions of BrC. BrC fractions are generally higher in the wet season than in the dry season. This is due to the enhanced formation of SOA and primary emissions from the canopy in comparison to the aerosol emissions from biomass burning [27]. During the wet season, the BrC varied between 10% in Alta Floresta and 27% in ATTO, while in the dry season, it varied between 9% and 25% for the same sites, respectively. These results are consistent with those obtained for the same sites in different studies but based on different methodologies. Ponczek et al., 2022 [28] showed that in Rio Branco, the BrC fraction was approximately 20% of the total absorption at 440 nm, and at the ATTO site, Saturno et al., 2018 [27] showed that the fraction was approximately 24% at 370 nm. It should be noted that both studies applied a similar methodology using in situ aerosol absorption measurements, which could introduce minor differences in the BrC fraction without compromising the overall comparison.

Figure 3 shows the spatial distribution of BrC and BC fractions, not discriminating them by season. Along the deforestation arc, the highest average fraction of BrC was observed in Cuiabá, at 15%, and the lowest in Alta Floresta, at 10%. Rio Branco and Ji-Paraná sites, approximately 1000 km apart, presented similar BC and BrC fractions. Sites in central Amazon showed BrC fractions up to twice as high as those in the deforestation arc, indicating the strong influence of the forest in maintaining the aerosol population in these regions. It is worth mentioning the similarity of the BrC fraction in Manaus and ATTO sites, and despite being located in the same region, they presented different emission patterns, one with urban and forest characteristics and the other exclusively forest. Although urban emissions may contribute to secondary organic aerosol formation downwind of Manaus, the forest is still the main source of these particles.

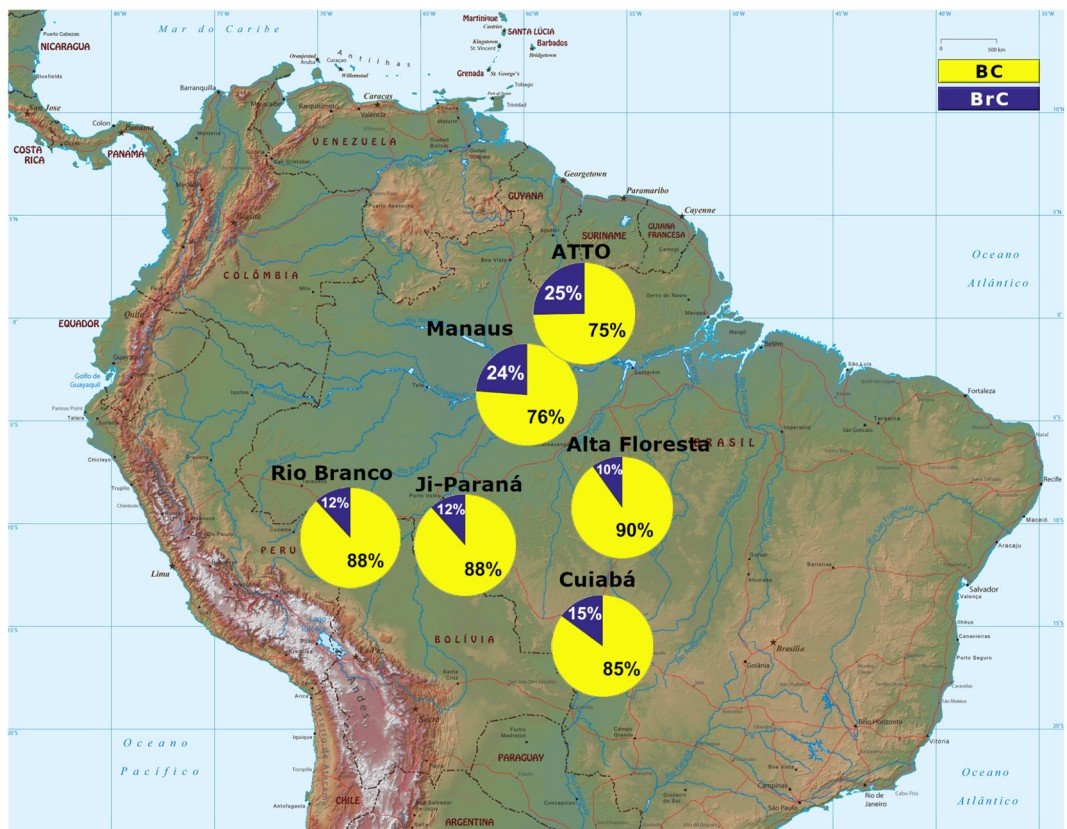

**Figure 3.** Spatial distribution of total BC and BrC fractions (here, considering the whole time series for each site), where in yellow is the portion of total BC and in blue the percentage of total BrC, obtained from the AAOD at 440 nm.

### 3.2. Carbonaceous Aerosol Mixture at the Amazonian Sites

Figure 4 shows the Ångström matrices for the sites along the deforestation arc. Each point represents a sample, colored by the month of occurrence. There is an evident similarity between the matrices. From August to December, which comprises the dry season, the BC contribution is dominant in all sites and agrees with the results presented in Table 2 and Figure 3. It should also be noted that all these sites did not have samples with a strong BrC contribution. To some extent, all sites along the deforestation arc have aerosol populations with mixtures of dust, BC, and BrC.

In Cuiabá and Alta Floresta, the BC/BrC mixture is more evident from August onwards, while in Rio Branco and Ji-Paraná, this aerosol population is also frequent from June onwards. Furthermore, Alta Floresta and Cuiabá had many samples classified as large particles mixed with BC, while in Ji-Paraná and Rio Branco, the frequency of this aerosol class was much lower. This is likely due to the long-range transport of Saharan and regional aerosol plumes that first reach Alta Floresta and Cuiabá and then go to the Ji-Paraná region and, to a lesser extent, Rio Branco [35].

It is interesting to note the presence of dust particles still in Rio Branco, located approximately 3500 km in a straight line from the Brazilian northeast coast. It shows how effective the long-range transport of aerosols by the trade winds is, which still has its influence even in the Brazilian northwest. It is also worth mentioning that Alta Floresta and Cuiabá had many samples classified as large particles or particles with low radiation absorption efficiency. According to Palácios et al., 2020 [24], they may be linked to the contribution of primary biogenic sources, which are more dominant during the wet season.

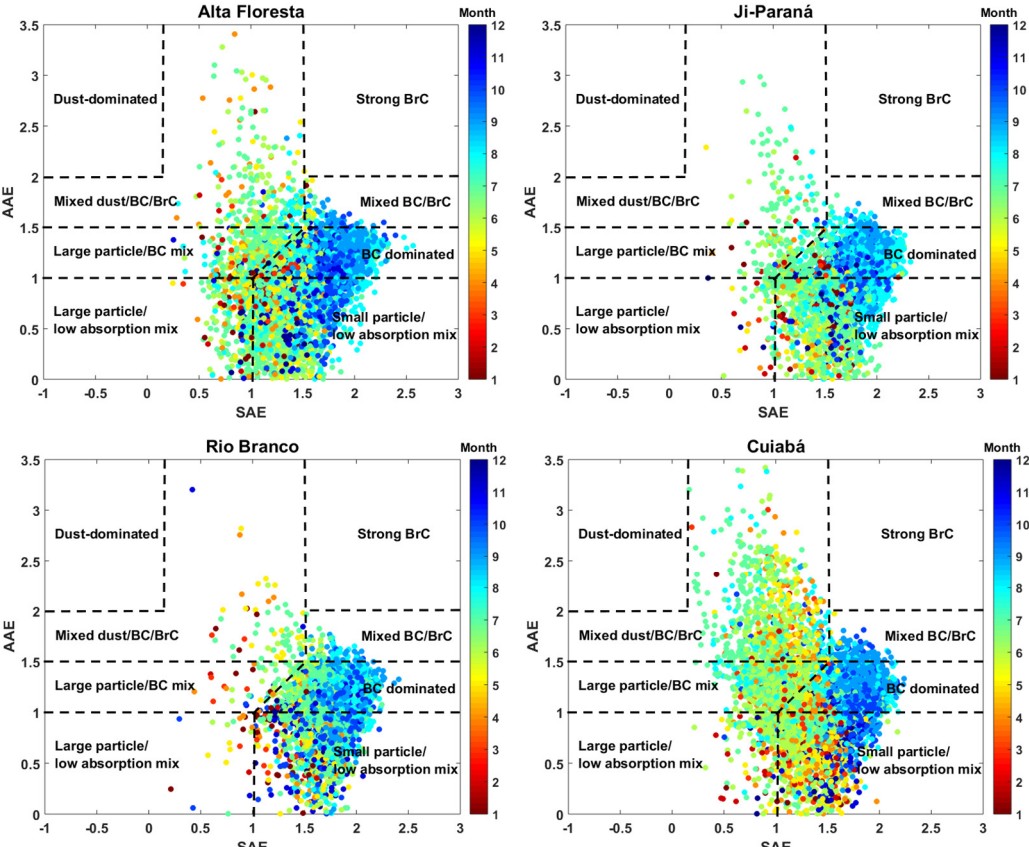

**Figure 4.** Ångström matrices were obtained with the AERONET network photometer data for the sites along the deforestation arc.

Figure 5 presents the Ångström matrices for the sites in central Amazon. ATTO and Manaus have lower data coverage than the deforestation arc sites since they suffer more from the presence of intense cloud cover and frequent rain. Although there is a limit to the extent of the analysis, it is possible to observe clear distinctions between aerosol populations. There are only a few dust-dominated points, but not exclusively BrC-dominated aerosol particles for the sites in central Amazon. The results show that, despite these sites receiving intense plumes from Africa, especially during the wet season, from January to April, aerosol populations are mainly composed of a mixture of dust, BC, and BrC, and aged particles from interactions with volatile organic compounds and other aerosols. It is important to emphasize that the low data coverage during the wet season can induce a bias regarding aerosol populations, especially in central Amazon sites.

Both sites present contributions from mixtures of large particles with low absorption properties, which are mainly related to primary biogenic emissions and are interestingly similar to Alta Floresta and Cuiabá. As expected, there is a clear BC dominance from August to December, but the frequency of observations from this aspect differs among the sites. It is worth noting that, in Manaus, the BC-dominated region presents more observations than the ATTO site, which may be linked to data coverage, but also likely due to urban emissions. Despite the minor differences, both sites look very similar regarding carbonaceous composition and can be even complementary to each other, which allows for assessing the completeness of the aerosol optical and chemical characteristics in central Amazonia.

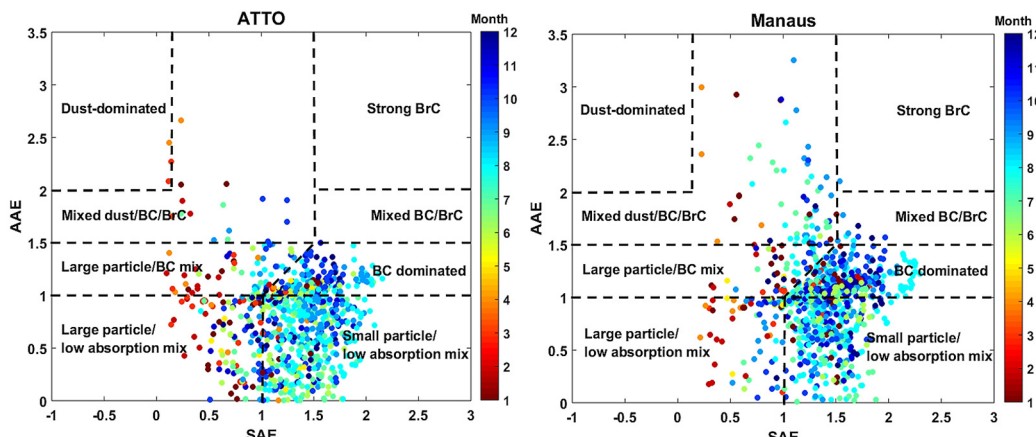

**Figure 5.** Ångström matrices obtained with sun photometer data of the AERONET network for the two sites in central Amazon.

### 3.3. Land Use and Cover at the Brazilian Amazonian AERONET Sites

The land use and cover of the Amazonian territory has been changing drastically throughout the last decades. Figure 6 presents the pattern of deforestation and the evolution of forest, agricultural, and other land-use classes for the Brazilian Amazon in two different years, 1985 and 2020. The pattern of deforestation follows mainly the railroads as fish spines geometries and is concentrated in the southern region of the Amazon Rainforest, where the deforestation arc is located. Approximately 89% of the Brazilian Amazon was classified as a rainforest in 1985, with 4.5% of it being agricultural land. In contrast, the forested area was reduced to approximately 78%, with 15% of agricultural land in 2020. Alta Floresta, Ji-Paraná, Rio Branco, and Cuiabá are cities in the deforestation arc, typically presenting a high number of fire outbreaks and the expansion of agricultural areas [24].

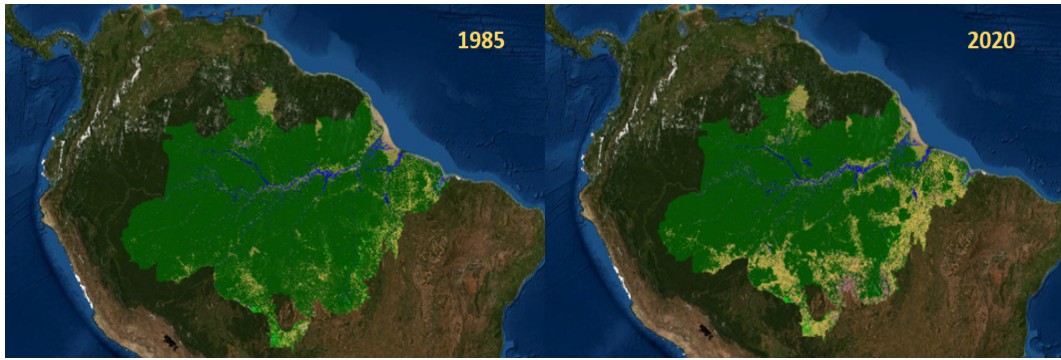

**Figure 6.** Land-use comparison between 1985 (**left**) and 2020 (**right**) for the legal Amazon. Deforestation and conversion of forest areas into agricultural and livestock regions is a strong mark—images obtained from the MapBiomas platform.

Table 3 presents the fraction of forested, agricultural, and non-forested natural formations in the municipality of the six Amazonian sites used in this study. For the ATTO site, we considered the municipality region of São Sebastião do Uatumã since the site is within this region, according to Andreae et al., 2015 [36]. In addition, Table 3 also presents the amount of forest loss and agricultural cover gain between 1985 and 2020. Alta Floresta is the site with the lowest fraction of forest and the highest fraction of agricultural land in 2020. In comparison, in 1985, the fraction of forest in Alta Floresta was approximately 85%, with 13% of agricultural land. This significant change was mainly induced by the gold rush in the 1980s and, more recently, the advance of different plantation crops, such as sugarcane and soybeans [37].

**Table 3.** Fraction of forested, agricultural, and non-forested natural formation in the six Amazonian sites in 2020 and the percentage of loss fraction of forest and agricultural land.

| Site | % Forest (2020) | % Loss$_{Forest}$ (1985–2020) | % Agricultural (2020) | % Gain$_{Agricultural}$ (1985–2020) | % Non-Forested Natural Formation (2020) |
|---|---|---|---|---|---|
| Alta Floresta | 46.59 | 38.92 | 51.73 | 38.45 | 0.23 |
| Ji-Paraná | 64.73 | 13.54 | 33.34 | 13.51 | 0.63 |
| Rio Branco | 67.62 | 21.62 | 30.62 | 21.10 | 1.08 |
| Cuiabá | 61.43 | 14.83 | 30.03 | 12.83 | 5.80 |
| Manaus | 80.44 | 0.35 | 3.14 | 0.21 | 0.48 |
| ATTO | 92.91 | 0.66 | 1.18 | 0.68 | 1.83 |

Following the same pattern, Ji-Paraná, Rio Branco, and Cuiabá also showed a strong reduction in forest cover, of 13.54, 21.62, and 14.83%, respectively, and an increase in the fraction of agricultural area of the same amount as the loss. The relationship is directly observed in Table 3. In 2020, these sites presented similar fractions of forest and agricultural land, although Cuiabá had a relatively large amount of non-forested natural formation.

In contrast, the sites in the central Amazon have the highest fraction of vegetation cover, and the lowest forest loss converted into agricultural areas. In 2020, Manaus presented 80.44% of forest cover, with a loss of 0.35% of the land to agricultural land over the last 35 years. São Sebastião do Uatumã, the region in which the ATTO site is located, is also one of the most preserved in the entire Amazon Rainforest [38]. In 2020, ATTO represented approximately 93% of forest cover and only 1.18% of the fraction dedicated to agriculture. The conversion of forest to agricultural areas was small, 0.66%.

The results show that the local atmospheric conditions in the ATTO surrounding areas are mainly dominated by forest emissions, as mentioned elsewhere [36]. Regarding the Manaus site, the region is also dominated by forest emissions [39–41]. However, the urban and thermoelectric power plant emissions may contribute to the atmospheric aerosol budget, which could drive, for instance, new organic aerosol particle formation.

Figure 7 shows the time series of forest and agricultural cover from 1985 to 2020 for each Amazonian site considered in this study. There is a direct relationship between the reduction in forests in contrast to the increase in the area dedicated to agriculture. Alta Floresta had the highest vegetation-to-agriculture conversion rate, particularly between 1985 and 2006. After government action and international pressure, deforestation was drastically reduced, and the areas have practically remained constant since then. Similar behavior is observed in Ji-Paraná, which presented a solid forest-agriculture conversion rate in the same period, with a drastic decrease in deforestation, influenced by the same public policies. It is important to note that although deforestation has been mitigated at these sites, secondary biomass burning may still represent the main source of aerosols into the atmosphere. This occurs mainly for the maintenance of the agricultural areas in those sites.

However, Rio Branco and Cuiabá did not show a reduction in the forest-agriculture conversion rate. They even maintained the trend of deforestation and a significant increase in the area converted to agriculture. A univariate linear regression model was applied to data from Cuiabá and Rio Branco to estimate the rate of forest-to-agriculture transformation over the years, whose results, consistent with a *p*-value < 0.05, are presented in Table 4. In the Cuiabá region, the forest loss rate was 1561 ± 61 ha year$^{-1}$, while the agricultural gain was 1367 ± 58 ha year$^{-1}$, which means that, if the tendency stays as it is, by 2053, the forest area will be equivalent to the area destined for agriculture.

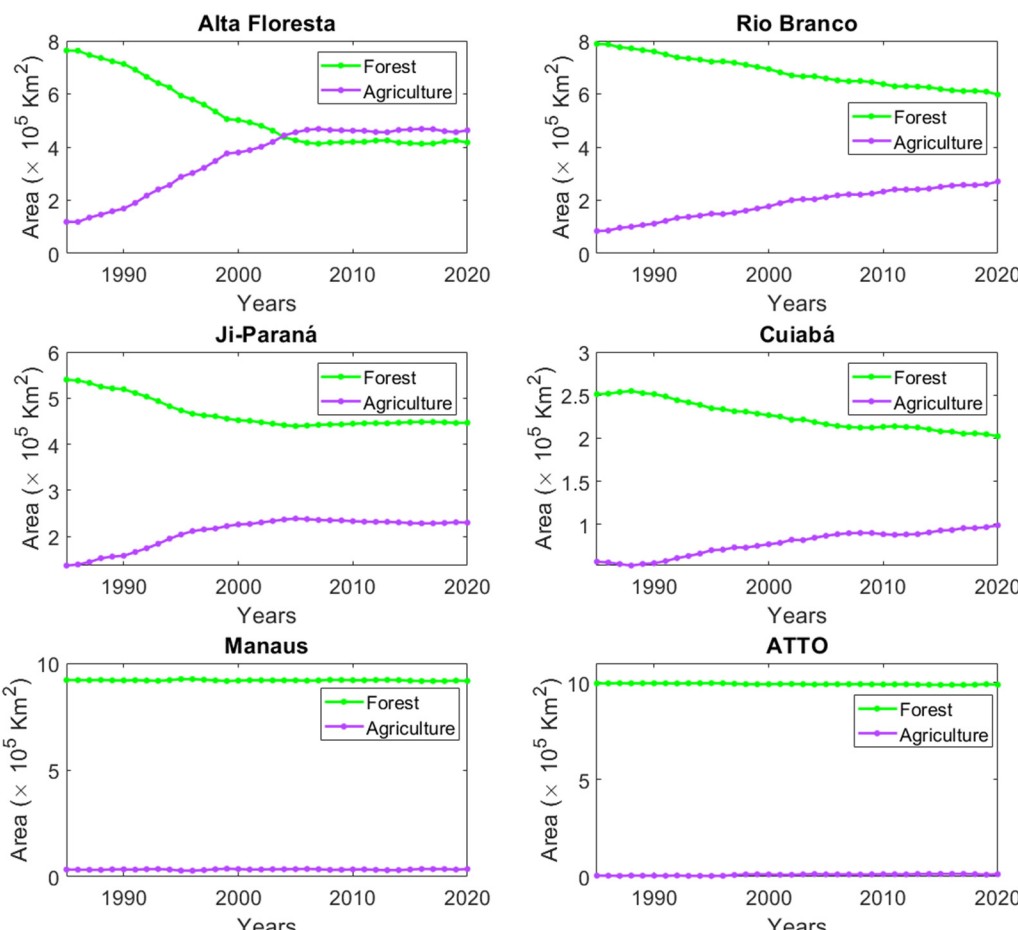

**Figure 7.** Time series of forest (green) and agriculture (magenta) areas for each municipal region of the sites investigated in this study. For the ATTO site, we used the urban region of São Sebastião do Uatumã, according to Andreae et al., 2015 [36]. Please note the different y-axis.

**Table 4.** Forest-to-agricultural zone conversion rates for Cuiabá and Rio Branco sites. Pearson's coefficient $R^2$ was obtained from the univariate linear fit with the data from each site.

| Site | Forest Loss Rate (ha year$^{-1}$) | $R^2$ | Agricultural Gain Rate (ha year$^{-1}$) | $R^2$ |
|---|---|---|---|---|
| Cuiabá | $1561 \pm 61$ | 0.95 | $1367 \pm 58$ | 0.94 |
| Rio Branco | $5530 \pm 134$ | 0.98 | $5413 \pm 134$ | 0.98 |

In approximately 125 years, the entire forest area surrounding Cuiaba will be converted into an agricultural zone. Rio Branco's situation is even more alarming since the forest-to-agriculture transformation rate is almost four times higher than in Cuiabá, at $5530 \pm 135$ ha year$^{-1}$ of forest loss and $5413 \pm 134$ ha year$^{-1}$ of agricultural gain. Thus, it is expected that by 2048, all the areas of forest and agriculture surrounding Rio Branco will be equivalent. Similar to Cuiabá, in approximately 125 years, all the forest cover would be converted into agricultural land. Note that the forest areas of Rio Branco and Cuiabá are not similar, being that the first one is approximately three times larger. The projections are based only on the rates obtained and do not consider legal limitations, such as the existence of integral protection units in these municipalities.

Sites located in central Amazon, in contrast, do not have a clear trend of reduction in forest areas or an increase in agricultural regions, as also shown in Table 3. In particular, these sites present their land use approximately constant over the years. However, there is a substantial risk that this pattern will change shortly due to the political management of the

Amazonian biome and the relaxation of environmental protection laws [42–44]. The change in land use, particularly related to deforestation, is an important driver of the observed changes in the region's rainfall regime and dry season length [31,45]. Saatchi et al. (2013) have shown that changes in forest cover lead to more occurrences of severe droughts in Amazonia, which is even more evident in the region of the deforestation arc [46]. Gatti et al. (2021) [47] argued that Amazonia is likely acting as a carbon source mainly due to deforestation, strongly intensifying the dry season and increasing fire occurrence. In addition, recent studies show that the forest has lost its resilience more quickly in forest regions most affected by droughts, fires, and urbanization [48].

*3.4. Connections between Aerosol Optical Properties, BC/BrC Fractions, and Land Use at the Brazilian Amazonian Sites*

The results show that sites with the highest forest-to-agriculture conversion rates have the highest AOD and total AAOD values, regardless of the season. This is associated with the deforestation process, which is mainly induced by primary biomass burning, and with the maintenance of agricultural areas by secondary fires. Alta Floresta and Ji-Paraná are two cases where the forest-to-agriculture conversion rate is approximately zero from 2005 onwards but still shows high values of AOD and total AAOD, mainly during the dry season, compared to the central Amazon sites. The high values of those aerosol optical properties are likely associated with the maintenance of agricultural regions. Rio Branco and Cuiabá, in fact, are the sites with the highest AAOD values in the wet and dry seasons, which also have positive and high conversion rates from forest to agriculture. At those sites, both primary and secondary biomass burning emissions dominate the aerosol loading in the atmosphere.

In contrast, despite receiving aerosol plumes from regional fires and long-distance transport, the central Amazon sites have the lowest AOD and AAOD values. In the wet season, the AOD is relatively higher at these sites, which mainly shows the effect of biogenic aerosols from the forest. This contribution is particularly observed in the Ångström matrices in Figure 5, where the region of biogenic aerosols, i.e., the region of large particles/low absorption mix, is more populated than the same region of the deforestation arc sites, shown in Figure 4. This effect is observed regardless of the season.

Figure 8 presents the relationships between land use and cover, i.e., forest and agricultural areas, and the total fractions of BrC and BC for each site. The relationship between the results presented in Table 1 and those in Table 4 and Figure 5 is even more evident. As suggested in Section 3.3, sites with the highest fraction of BC also have the most significant agricultural areas. That is, they currently have the lowest fraction of forest compared to more preserved sites, such as sites in central Amazonia. In contrast, the sites with a high fraction of forest cover have the highest total fraction of BrC, which also evidences the role and importance of the forest in the maintenance of organic aerosols.

Although the selected sites were located far away from each other, Figure 3 shows that the BC and BrC fractions for Rio Branco, Ji-Paraná, and Cuiabá are practically similar. Furthermore, the fractions of forest and agricultural areas are similar for these three sites, as shown in Table 3. The results indicate that the type of land use and the processes by which land is transformed reflect the population of carbonaceous aerosols in the atmosphere. In particular, regions that suffer the most from deforestation have higher fractions of BC, while those with more significant forest cover have a higher fraction of BrC.

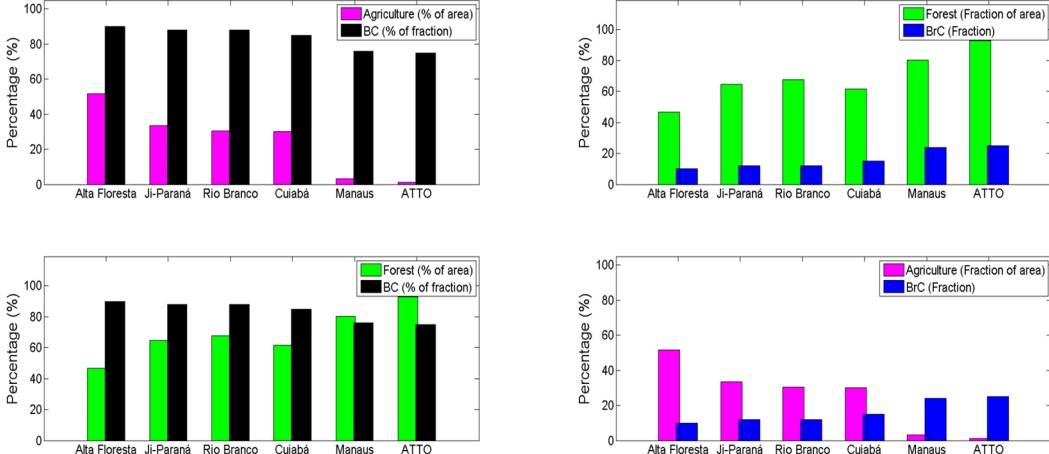

**Figure 8.** Bar plots of BC and BrC fractions compared to the forest and agricultural land fractions for the six Amazonian sites.

## 4. Conclusions

The present study shows for the first time the spatial distribution of BC and BrC light-absorbing aerosols in six sites in the Amazon Rainforest and their associations with land use. Results showed that areas with forest loss were mostly converted into agricultural areas at all sites. Alta Floresta was the site with the highest forest loss (39%) from 1985 to 2020, followed by Rio Branco (22%). In contrast, the ATTO site region has the most preserved forest area, with approximately 93% of the original vegetation. AOD and AAOD showed a marked seasonality with higher values in the dry season because of the influence of biomass burning emissions. During the wet season, AOD (500 nm) was similar among the sites, ranging from 0.08 in Alta Floresta to 0.13 at the ATTO site. Interestingly, the mean AOD and AAOD were higher in the ATTO forest site compared to the Alta Floresta site during the wet season, indicating the relevance of biogenic emissions to sustain the aerosol population in the forest. In contrast, the spatial differences were larger during the dry season, with AOD values ranging from 0.19 at the ATTO site to 0.49 in Alta Floresta. The impact of biomass burning emissions on AAOD values in the deforestation arc was evident, with higher values in Rio Branco and Cuiabá compared to the Manaus and ATTO forest sites.

The contribution of BC to AAOD was greater at the sites along the deforestation arc, with values approximately three times higher compared to the sites in central Amazon. During the wet season, the BrC contribution to AAOD varied between 10% in Alta Floresta and 27% in ATTO, while in the dry season, it varied between 9% and 25% in the same sites, respectively. These results strongly indicate that sites heavily impacted by forest-agriculture transformation processes also have the highest average BC contribution to AAOD. Conversely, sites with a higher fraction of forest cover have the highest fractions of BrC.

Ångström matrices showed that all sites along the deforestation arc have aerosol populations with mixtures of BC and BrC. The presence of dust mixed with BC and BrC aerosols was observed even on the western site of Rio Branco, indicating the influence of long-range transport of soil dust across the Amazon Rainforest. The central Amazonian sites showed minor differences between them, with a greater contribution of BC-dominated aerosols in Manaus compared to ATTO.

Therefore, the results showed that Amazonian regions with different degrees of land-use transformation have different proportions of light-absorbing carbonaceous aerosols. This implies differences in the atmospheric radiative balance of each region, with possible direct and indirect effects on microclimate and mesoscale atmospheric dynamics. Future studies are required to evaluate the impacts of different types of biomass burning emissions

(primary and secondary) and transport to define to what extent these factors are significant to the BC and BrC aerosol fraction.

**Author Contributions:** F.G.M. and M.A.F. equally contributed to this study. Conceptualization, F.G.M., M.A.F. and P.A.; methodology, F.G.M., M.A.F. and H.M.J.B.; analysis, M.A.F., H.M.J.B. and R.P.; validation, M.A.F., F.G.M. and H.M.J.B.; formal analysis, F.G.M. and M.A.F.; investigation, F.G.M. and M.A.F.; resources, P.A., H.M.J.B., F.J., J.S.S., B.N.H., E.L., L.V.R. and L.A.T.M.; data curation, M.A.F. and H.M.J.B.; writing—original draft preparation, F.G.M., M.A.F. and R.P.; writing—review and editing, all authors; supervision, P.A., H.M.J.B., L.A.T.M. and E.L.; project administration, P.A.; funding acquisition, P.A. All authors have read and agreed to the published version of the manuscript.

**Funding:** This study was supported by Fundação de Amparo à Pesquisa do Estado de São Paulo FAPESP, projects 2017/17047-0 and 2021/13610-8.

**Institutional Review Board Statement:** Not applicable.

**Informed Consent Statement:** Not applicable.

**Data Availability Statement:** The AERONET website provides data analysis and dissemination tools at https://aeronet.gsfc.nasa.gov (accessed on 1 June 2022). Data can be viewed in charts using the data display interface, acquired using the data download tool, analyzed, and downloaded using the analysis tools provided by AERONET. Land-use data were obtained through the MapBiomas platform at https://mapbiomas.org/ (accessed on 1 June 2022).

**Acknowledgments:** The authors thank the field researchers and technicians Delano Campos, Bruno Takeshi, Edilson Andrade, Alberto W. Dresch, João Basso, Paulo Arruda, and Alejandro Fonseca Duarte for maintaining and operating the NASA/AERONET network over so many years.

**Conflicts of Interest:** The authors declare no conflict of interest.

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
