# Peer review of "Relationship between Land Use and Spatial Variability of Atmospheric Brown Carbon and Black Carbon Aerosols in Amazonia"

_atmosphere, doi:10.3390/atmos13081328_

Round 1

Reviewer 1 Report

This paper reports on a study of the relationship between land use and spatial variability of atmospheric brown carbon and black carbon aerosols in Amazonia. The fraction and seasonality of the black carbon (BC) and brown carbon (BrC) contributions to absorption at 440 nm were estimated by using AERONET long-term measurements of aerosol optical depth (AOD) in 500 nm and absorption AOD (AAOD) in 440, 675, and 870 nm. Land use data from the MapBiomas collection 6.0 was used to access the land transformation from forest to agricultural areas on each of these sites. The results show for the first time important geographical and seasonal variability of the aerosol optical properties, particularly the BC and BrC contributions. The effects of the land use change could introduce differences in the radiative balance in the different Amazonian regions. Therefore, it is a topic of interest to the researchers in the related areas and the results show that synthesizing these analyses allows a better understanding of the role of emissions from the Amazon rainforest that could have global impacts.

In my opinion, this manuscript can be accepted after revising the manuscript. My detailed comments are as follows: 

1.       Many years and seasons are involved in the manuscript and need to be clarified in the text during the analysis. For example, the data presented in Tables 1 and 2 lack a description of specific years. Are the conclusions for lines 201-204 obtained under the dry season? It can be seen from Table 1 that there is no such phenomenon in the wet season. In lines 420-421, the forest-to-agriculture conversion rate is approximately zero in Alta Floresta and Ji-Paraná. When was the exact year that this conclusion was reached? The authors should check the manuscript carefully. The explanation should be added to the manuscript.

2.       Line 17: Do the parameters 500, 440, 675, and 870nm given here refer to the wavelength? Besides, how did the authors determine these values in this manuscript? In order to explain more clearly, the authors should give some descriptions and reasons.

3.       As mentioned in Lines 149-150, this study applied the same methodology as Cazorla et al., 2013 [17] for different Amazonian regions associated with different types of aerosol populations. How did the authors determine the correctness of using this method under the conditions of this manuscript?

4.       Lines 135-136: The refractive index and density of BC are given here. Are these assumed to be consistent at given different wavelengths?

5.       Line 16: The first occurrence of “AERONET” should be given the full name.

6.       Line 124: “composed by” should be corrected as “composed of”.

7.       Line 163Does the appearance of the peak in Figure 1 represent June?

8.       Line 319: Why are ATTO and Manaus complementary to each other in some ways?

Reviewer 2 Report

The study by Morais et al. tries to investigate the long-term variations in optical properties of aerosols across Amazonian regions. The authors combined AERONET measurements with MapBiomas platform data of land use to see how BC and BrC have changed over time with respect to deforestation. It was concluded that forest regions experienced more BrC than BC while land use changes (to agricultural areas) have been associated with more BC than BrC. In general, the work is fine given that a long-term dataset has been investigated which shed light on the BC/BrC variations over time. In my view, the work can be published in the atmosphere journal after addressing the following comments:

Comments:

1) Lines 13 and 37: These sentences are exactly the same in Abstract and Introduction. It would be better to rephrase one of them.

 2) Lines 54-58: Along with SOA and biological processes, brown carbon can also be emitted from biomass burning and fuel combustion. These are also important sources of BrC and should be included in this paragraph.

doi.org/10.1021/acs.estlett.8b00118

doi.org/10.1016/j.scitotenv.2019.135902

doi.org/10.5194/acp-20-2017-2020

 3) Lines 63 to 69: This paragraph intends to show how AERONET measurements can be helpful in characterizing aerosol properties. However, it needs to at least discuss a few case studies that used such measurements:

doi.org/10.5194/acp-14-12271-2014

doi.org/10.1029/2004JD005274

 4) Lines 167-173: How about the impact of aerosol wet deposition on AOD during Jan-Jun?

 5) Figure 2 and lines 218-220: Lower AAOD values specifically for BC were observed during 2012 to 2017. Is it associated with meteorological reasons?

 6) Lines 248-249: What is the reason for higher BrC in wet season? Higher emissions? Or enhanced formation of SOA?

 7) Line 327: The “deforestation arc” has been mentioned several times throughout the manuscript while finally being defined in this part. It is better to provide the corresponding details earlier.

 8) “Despite being sites …” is grammatically wrong. Replace it with “Although the selected sites were located far away …”

9) The conclusion should be written more concisely. No need to mention the methodology and detailed results.
